# PU.1 Expression Defines Distinct Functional Activities in the Phenotypic HSC Compartment of a Murine Inflammatory Stress Model

**DOI:** 10.3390/cells11040680

**Published:** 2022-02-15

**Authors:** James S. Chavez, Jennifer L. Rabe, Giovanny Hernandez, Taylor S. Mills, Katia E. Niño, Pavel Davizon-Castillo, Eric M. Pietras

**Affiliations:** 1Division of Hematology, Anschutz Medical Campus, University of Colorado, Aurora, CO 80045, USA; james.chavez@cuanschutz.edu (J.S.C.); rabejenn@gmail.com (J.L.R.); gioh300@gmail.com (G.H.); taylor.mills@cuanschutz.edu (T.S.M.); katia.nino@cuanschutz.edu (K.E.N.); 2Department of Pediatrics, Anschutz Medical Campus, University of Colorado, Aurora, CO 80045, USA; pavel.davizon-castillo@coloradochildrens.org; 3Department of Immunology and Microbiology, Anschutz Medical Campus, University of Colorado, Aurora, CO 80045, USA

**Keywords:** hematopoiesis, inflammation, hematopoietic stem cell, PU.1, myeloid, megakaryocyte

## Abstract

The transcription factor PU.1 is a critical regulator of lineage fate in blood-forming hematopoietic stem cells (HSC). In response to pro-inflammatory signals, such as the cytokine IL-1β, PU.1 expression is increased in HSC and is associated with myeloid lineage expansion. To address potential functional heterogeneities arising in the phenotypic HSC compartment due to changes in PU.1 expression, here, we fractionated phenotypic HSC in mice using the SLAM surface marker code in conjunction with PU.1 expression levels, using the *PU.1-EYFP* reporter mouse strain. While PU.1^lo^ SLAM cells contain extensive long-term repopulating activity and a molecular signature corresponding to HSC activity at steady state, following IL-1β treatment, HSC^LT^ induce PU.1 expression and are replaced in the PU.1^lo^ SLAM fraction by CD41^+^ HSC-like megakaryocytic progenitors (SL-MkP) with limited long-term engraftment capacity. On the other hand, the PU.1^hi^ SLAM fraction exhibits extensive myeloid lineage priming and clonogenic activity and expands rapidly in response to IL-1β. Furthermore, we show that EPCR expression, but not CD150 expression, can distinguish HSC^LT^ and SL-MkP under inflammatory conditions. Altogether, our data provide insights into the dynamic regulation of PU.1 and identify how PU.1 levels are linked to HSC fate in steady state and inflammatory stress conditions.

## 1. Introduction

Blood system responses to physiological stress require a coordinated and dynamic re-balancing of hematopoietic stem cell (HSC) fate to meet changes in demand for specific blood lineages while preserving long-term HSC function and lifelong blood system maintenance [1]. HSC can respond to numerous inflammatory signals, including direct sensing of pathogen components, endogenous danger signals, and/or a wide variety of cytokines, including interleukin-1β (IL-1β), which is produced in response to many physiological stresses [2]. IL-1β induces numerous changes in the blood system, including increased production of myeloid cells and platelets to facilitate host defense and tissue repair [3,4].

Previously, we and others have shown that IL-1β activates expression of the transcription factor PU.1, both at the RNA and protein levels, via direct IL-1R signaling in HSC [3,5,6,7,8,9]. PU.1/Spi1 is an ETS-family transcription factor that serves as a ‘master regulator’ of blood lineage output [10]. Levels of PU.1 expression are lineage-specific, with the highest levels expressed in myeloid-committed cells, intermediate levels in lymphocytes, and the lowest levels in megakaryocyte (Mk)- and erythroid-lineage cells [11,12]. HSC themselves have been shown to express very low levels of PU.1, which is associated with homeostatic blood lineage output and normal self-renewal function [10,13,14]. On the other hand, the induction of PU.1 in HSC by IL-1β and other inflammatory cytokines is associated with chronic overproduction of myeloid cells and platelets in vivo [3]. Recently, we found that increased PU.1 expression in long-term HSC (HSC^LT^) also rapidly engages a quiescence-enforcing program that represses protein synthesis and cell cycle genes, thereby preventing spurious proliferation and maintaining HSC pool size in response to inflammatory stress [7]. Hence, PU.1 appears to modulate multiple fate choices that impact blood production in response to inflammatory stress.

While these findings provide important insight into HSC^LT^ function, the extent to which PU.1 is heterogeneously expressed within the phenotypic HSC compartment under homeostatic and inflammatory stress conditions has not to our knowledge been investigated in vivo. Furthermore, how PU.1 levels within the HSC compartment facilitate the diverse cell fate outcomes required to maintain increased myeloid cell and platelet output, while preserving long-term HSC function, has not been clearly established. Here, we use the *PU.1-EYFP* reporter mouse strain to identify and characterize the impact of low and high PU.1 expression levels on cells in the HSC-enriched SLAM compartment (Lin^−^/c-Kit^+^/Sca-1^+^/Flk2^−^/CD48^−^/CD150^+^) under homeostatic conditions and in response to acute and chronic IL-1β stimulation. We find that PU.1 levels are dynamically regulated in the SLAM compartment in response to inflammatory stress, and PU.1 expression is associated with self-renewal capacity and patterns of blood lineage priming and output in SLAM cells. Lastly, we show that in response to inflammatory stress, the PU.1^lo^ SLAM fraction becomes enriched for stem-cell-like megakaryocyte-committed progenitors (SL-MkP) that serve as an ‘emergency’ reservoir for platelet production. We also show that EPCR and CD41, but not CD150, can robustly separate HSC^LT^ and SL-MkP in this setting. Taken together, the present study links PU.1 levels to distinct cell fate outcomes in response to acute and chronic inflammation.

## 2. Materials and Methods

### 2.1. Mice

Wild-type C57BL/6 and congenic B6.SJL-*Ptprc^a^Pepc^b^*/BoyJ (Boy/J) mice were obtained from the Jackson Laboratory. *PU.1-EYFP* mice [15,16] were a kind gift of Dr. Claus Nerlov (MRC Weatherall Institute). All animal procedures were approved by the University of Colorado Anschutz Medical Campus Institutional Animal Care and Use Committee (IACUC), protocol #00091. Mice were housed in a temperature- and light-controlled facility in HEPA-filtered cages and provided with chow and water ad libitum.

### 2.2. Flow Cytometry

Bone marrow harvest and flow cytometry were performed as previously described [7]. For sorting experiments, BM cells were isolated from the arms, legs, pelvis, and spines by crushing bones in a mortar with a pestle containing staining media (SM—HBSS supplemented with 2% fetal bovine serum (FBS)) and transferred over a 70-micron filter. Cells were resuspended in ACK for five minutes, incubated on ice, and washed with SM. Erythrocyte-depleted BM cells were resuspended in 3 mL SM and transferred over a 70-micron filter. A ficoll gradient was created to separate out cellular debris by adding 2 mL Histopaque 1119 (Sigma-Aldrich, St. Louis, MO, USA) to the bottom of the cell suspension followed by centrifugation at 1200 RPM (Beckman Coulter, Brea, CA, USA) with the break turned off. The top liquid phase containing BM cells was transferred into a new tube and washed once with SM. BM cells were resuspended in a solution of 100 µL SM and 5 µL murine CD117 microbeads (Miltenyi Biotec, Auburn, CA, USA) per mouse, incubated on ice for 20 min, washed, and enriched on an autoMACS Pro separator (Miltenyi Biotec). For immunophenotyping analyses, femurs and tibiae were flushed using a syringe containing 3 mL SM and affixed with a 1 ½ inch × 21-gauge needle. BM cells were erythrocyte-depleted by ACK treatment, resuspended in SM, and counted using a ViCell automated cell-counter (Beckman Coulter, Brea, CA, USA). For HSPC analyses, BM cells were blocked with purified Rat IgG (Sigma-Aldrich, St. Louis, MO, USA) and stained with an antibody cocktail in SM containing CD3 PE/Cy5 (eBioscience, San Diego, CA, USA, 15-0031-81), CD4 PE/Cy5 (eBioscience, 15-0041-82), CD5 PE/Cy5 (Biolegend, San Diego, CA, USA, 100610), CD8 PE/Cy5 (Biolegend, San Diego, CA, USA, 100710), B220 PE/Cy5 (Biolegend, 103210), Ter119 PE/Cy5 (Biolegend, 116210), Gr1 PE/Cy5 (Biolegend, 108410), EPCR PE (eBioscience, 12-2012-82), CD11b PE/Cy7 (Biolegend, 101216), CD34 AF647 (BD Biosciences, 560230), CD48 A700 (Biolegend, 103426), c-Kit APC/Cy7 (Biolegend, 105826), and Flk2 Biotin (Biolegend, 135308) and incubated for 30 min on ice. Subsequently, cells were stained with an antibody cocktail in SM containing Brilliant Buffer (BD Biosciences), Sca1 Bv421 (Biolegend, 108128), CD41 BV510 (Biolegend, 113923), Streptavidin-BV605 (BD Biosciences, 563260), CD16/32 BV711 (Biolegend, 101337), and CD150 BV785 (Biolegend, 115937) for 30 min on ice. Cells were resuspended in a solution of SM containing propidium iodide (PI) for dead cell exclusion. For mature BM cell analyses to monitor BM chimerism, peripheral blood cells were RBC-depleted using ACK as above and stained with an antibody cocktail of CD8-PE (Biolegend, 100708), Mac-1 PE/Cy7, IgM APC (eBioscience, 17-1590-82), CD3 AF700 (Biolegend, 100216), CD19 APC/Cy7 (Biolegend, 115530), Gr1 PB (Biolegend, 108430), CD4 BV510 (Biolegend, 100449), Ly6C BV605 (BD Biosciences, 563011), and B220 BV875 (Biolegend, 103246) and incubated for 30 min on ice. Cells were washed and resuspended in PI. For liquid culture flow analyses, cells were stained with an antibody cocktail containing CD9 PE (Biolegend, 124805), Sca1 PE/Cy7 (Biolegend, 108114), CD11b APC (Biolegend, 101212), c-Kit APC/Cy7, Gr1 PB, CD41 BV510, CD16/32 BV711, and CD150 BV785 (catalog numbers same as listed above) and incubated on ice for 30 min. The cells were washed and resuspended in DPBS with PI. Data were acquired on a 12-channel, 3-laser FACSCelesta or an 18-channel, 5-laser LSRFortessa (Becton, Dickenson and Co., Franklin Lakes, NJ, USA) and analyzed using FlowJo v10 (FlowJo LLC, Ashland, OR, USA).

### 2.3. In Vivo Procedures

In vivo IL-1β stimulation was performed as previously described [7]. Recombinant murine IL-1β was resuspended in sterile D-PBS/0.2% BSA and injected in a 100 μL bolus of 0.5 μg once daily via the intraperitoneal (i.p.) route, using a 31 g insulin syringe. Transplantation experiments were performed as previously described [6]. CD45.1^+^ Boy/J recipient mice were lethally irradiated (11 Gy, split dose 2 h apart) using a cesium source (JL Shepherd & Associates, San Fernando, CA, USA) and maintained on Bactrim in autoclaved water for 4 weeks post transplant. Mice were injected retro-orbitally with 250 sorted SLAM cells (either PU.1^hi^ or PU.1^lo^) isolated from donor mice along with 5 × 10^5^ Boy/J helper cells that were Sca-1-depleted using an AutoMACS (Miltenyi Biotec, Auburn, CA, USA) and resuspended in SM. Mice were bled retro-orbitally every four weeks for 20 weeks to monitor donor chimerism. Blood collection was performed via retro-orbital route into EDTA-coated microtainers. Complete blood count (CBC) analyses were performed on a calibrated Element HT5 (Heska, Loveland, CO, USA) hematology analyzer.

### 2.4. Cell Culture

All cell cultures were performed in a humidified incubator (Thermo Fisher Scientific, Waltham, MA, USA) at 37 °C and 5% CO_2_. For liquid and methylcellulose cultures, PU.1 eYFP low and high SLAM cells were purified from mice that were treated with either PBS or 0.5 µg IL-1β (Peprotech, Rocky Hill, NJ, USA) 24 h prior to harvest. PU.1 low and high were selected based on median fluorescent intensity (MFI), where approximately the lowest 25% were designated PU.1 low and the highest 25% designated as PU.1 high. Pools of 400 cells per well were sorted for liquid culture into Stempro 34 medium (Gibco, Waltham, MA, USA, 10639011) and supplemented with antibiotic–antimycotic (Gibco, 15240062), 2 mM l-glutamine (Gibco, 25030024), 25 ng/mL IL-11 (Peprotech), 25 ng/mL SCF (Peprotech), 25 ng/mL TPO (Peprotech), 25 ng/mL Flt3L (Peprotech), 10 ng/mL IL-3 (Peprotech), 10 ng/mL GM-CSF (Peprotech), and 4 U/mL EPO (Peprotech). Half of the medium was replenished every two days, and cells were harvested for flow cytometry analysis and cell counts on days 4 and 8. Methylcellulose cultures were executed using Methocult M3231 Base Media (Stemcell Technologies, Cambridge, MA, USA 03231) containing antibiotic–antimycotic, IMDM, 25 ng/mL IL-11, 25 ng/mL SCF, 25 ng/mL TPO, 25 ng/mL Flt3L, 10 ng/mL IL-3, 10 ng/mL GM-CSF, and 4 U/mL EPO. Colonies were counted and phenotyped at day 8.

### 2.5. Fluidigm qRT-PCR Analysis

Fluidigm qRT-PCR analysis was performed as previously described [7]. Briefly, 100 PU.1 low and PU.1 high SLAM cells per well were sorted directly into 5 µL of 2× Reaction Mix (Invitrogen, Waltham, MA, USA) contained within a 96-well PCR plate. Once all cells were sorted, the plates were sealed with an aluminum seal, spun in a centrifuge at 1200 RPM for 5 min, snap frozen in liquid nitrogen, and stored at −80 °C. cDNA was generated from RNA using Superscript III (Invitrogen) and a custom primer set mix (Fluidigm, San Francisco, CA, USA) that were amplified for 18 cycles on a thermocycler (Eppendorf, Hamburg, Germany). Excess primers were removed from the samples by Exonuclease I (NEB, Ipswich, MA, USA) treatment, and the samples were diluted in DNA suspension buffer (Teknova, Hollister, CA, USA). Pre-amplified cDNA and custom primer sets were loaded onto a Fluidigm 96.96 Dynamic Gene Expression IFC. Subsequently, the IFC was run on a Biomark HD (Fluidigm) with SsoFast Sybr Green (Bio-Rad, Hercules, CA, USA) used for detection. Fluidigm gene expression software was used to analyze the data, and all values are relative to *Gusb.* Relative changes in gene expression were determined using the ΔΔCT approach.

### 2.6. Statistical Analysis

Statistical analyses were performed using Prism (GraphPad, San Diego, CA, USA). Multivariate comparisons were made using ANOVA with Tukey’s post-test. *p*-values < 0.05 were considered statistically significant. For in vivo studies, groups of mice receiving treatments (IL-1, transplant) were chosen randomly within each cage. Experiments were repeated at least twice to ensure reproducibility of findings.

## 3. Results

### 3.1. PU.1 Levels Identify Distinct Functional Activities in the SLAM Compartment following Chronic IL-1β Treatment

We previously showed that chronic IL-1β treatment increases PU.1 levels in HSC and is associated with precocious myeloid differentiation [3]. Hence, we reasoned that under inflammatory conditions, increased PU.1 expression would impair HSC self-renewal, whereas low PU.1 levels would mark immature long-term HSC. To test our hypothesis, we treated *PU.1-EYFP* reporter mice with IL-1β or PBS (hereafter referred to as -IL-1β) daily for 20 days to simulate a chronic inflammatory state. The *PU.1-EYFP* reporter mouse line harbors EYFP fused to the C-terminus of PU.1 and is knocked into the endogenous *PU.1* locus, allowing for prospective isolation of cells based on PU.1 protein levels [3,7,8,9,15,17]. Following IL-1β treatment, we verified increased numbers of neutrophils and platelets in the peripheral blood by CBC, and as we previously reported, we noted increased overall numbers of phenotypic HSC-enriched SLAM cells in the BM (Appendix A) [6,7]. Within the SLAM cell population, we found an increased proportion and absolute number of PU.1^hi^ SLAM cells following IL-1β treatment (Figure 1A,B and Appendix A). We then sorted PU.1^lo^ and PU.1^hi^ SLAM cells based on the upper and lower quadrants of PU.1-EYFP expression in -IL-1β controls and assessed short-term potential using methylcellulose colony-forming unit (CFU) assays to read out myeloid colony-forming activity. Interestingly, we noticed that PU.1^lo^ SLAM cells from IL-1β-treated mice exhibited significantly reduced clonogenic activity in this setting (Figure 1C). On the other hand, clonogenic activity was more abundant in PU.1^hi^ SLAM cells irrespective of IL-1β treatment, indicative of increased myeloid potential in these fractions.

To measure long-term reconstitution activity in these SLAM cell fractions, we purified and transplanted 250 PU.1^lo^ or PU.1^hi^ SLAM cells from PU.1-EYFP mice treated ± IL-1β into lethally irradiated recipient mice and then analyzed post-transplant donor-derived myeloid/lymphoid chimerism in the peripheral blood every 4 weeks for 20 weeks (Figure 1D). While PU.1^lo^ SLAM cells from -IL-1β donor mice exhibited robust long-term reconstitution activity as we had anticipated, unexpectedly, the PU.1^lo^ SLAM fraction from IL-1β-treated donor mice did not contribute significantly to the peripheral blood (Figure 1E), and what few donor cells were present did not exhibit a consistent phenotypic trend relative to -IL-1β PU.1^lo^ controls toward myeloid or lymphoid hematopoiesis (Appendix A). Limited reconstitution activity in the PU.1^lo^ SLAM fraction was further reflected by the lack of donor contribution to the SLAM cell compartment in the recipient mice BM after 20 weeks (Figure 1F). On the other hand, the PU.1^hi^ SLAM fractions from mice treated ± IL-1β provided an intermediate level of reconstitution activity, with PU.1^hi^ SLAM cells from IL-1β-treated donor mice providing a more stable long-term contribution to the blood (Figure 1E), though their contribution to the BM SLAM compartment was more variable than the -IL-1β PU.1^hi^ controls (Figure 1F). Furthermore, we did not see evidence of overt ‘myeloid bias’ associated with high PU.1 expression (Appendix A). Instead, PU.1^hi^ SLAM fractions from IL-1β-treated donor mice exhibited increased lymphoid reconstitution activity and decreased myeloid activity relative to their PU.1^lo^ counterparts. Taken together, we identified unexpected patterns of long-term reconstitution in PU.1^lo^ and PU.1^hi^ SLAM cells, with the PU.1^lo^ fraction from IL-1β-treated mice exhibiting essentially no long-term reconstitution activity.

### 3.2. EPCR^+^ HSC^LT^ Are Depleted from the PU.1^lo^ SLAM Fraction following IL-1β Treatment

Given that PU.1 serves as a myeloid/lymphoid differentiation factor, we had originally anticipated that the PU.1^lo^ SLAM compartment would retain extensive HSC potential regardless of IL-1β stimulation. Since our transplantation assay showed a loss of repopulating activity in this fraction following IL-1β treatment, we isolated PU.1^lo^ and PU.1^hi^ SLAM cells and analyzed the expression of genes related to HSC self-renewal (Figure 2A). Interestingly, PU.1^lo^ SLAM cells from IL-1β-treated mice expressed significantly lower levels of self-renewal-associated genes *Hoxb5* [18,19] and *Egr1* [20], suggesting that HSC in this compartment either were downregulating these genes or were being depleted from the PU.1^lo^ SLAM fraction (Figure 2B). To address this question, we analyzed the surface phenotype of PU.1^lo^ and PU.1^hi^ SLAM cells using EPCR and CD34. We previously demonstrated that EPCR^+^/CD34^−^ SLAM cells contain the majority of long-term HSC (HSC^LT^) activity, and these markers faithfully enrich for HSC^LT^ within the SLAM compartment of IL-1β-treated mice [6,7]. Following IL-1β treatment, we observed an expected decrease in the frequency of phenotypic EPCR^+^/CD34^−^ HSC^LT^ (Appendix A), but as previously published, this is due to the expansion of EPCR- cells in the SLAM compartment (which contributes to the overall increase in SLAM cells (Appendix A) we observe following IL-1β treatment) rather than HSC^LT^ depletion. Interestingly, whereas the PU.1^lo^ SLAM cell fraction from -IL-1β control mice contained a high frequency of phenotypic EPCR^+^/CD34^−^ HSC^LT^, these cells were essentially absent in the PU.1^lo^ SLAM cell fraction of IL-1β-treated mice (Figure 2C,D). On the other hand, the PU.1^hi^ SLAM fraction retained a similar frequency of phenotypic EPCR^+^/CD34^−^ HSC^LT^ following IL-1β treatment (Figure 2C,D). Consistent with these findings, we noticed that PU.1 levels increased significantly following IL-1β treatment in EPCR^+^ SLAM cell fractions but not in EPCR^−^ SLAM cells (Appendix A). Hence, EPCR^+^ HSC^LT^ are no longer prospectively enriched by low PU.1-EYFP levels. Lastly, we confirmed that EPCR expression enriches for HSC at the molecular level by qRT-PCR (Appendix A). EPCR^+^ SLAM cells expressed higher levels of *Bmi1* and *Hoxb5* regardless of IL-1β exposure, in line with our previously published data [6]. Altogether, our data show that increased PU.1 expression in response to IL-1β treatment occurs exclusively in EPCR^+^ SLAM cells, leaving only EPCR^−^ SLAM cells with limited HSC gene expression levels in the PU.1^lo^ phenotypic gate.

### 3.3. Distinct Patterns of Lineage Priming in the SLAM Compartment Associated with PU.1 Levels

We next sought to characterize the cells remaining in the PU.1^lo^ SLAM fraction following IL-1β treatment. Given the reduced expression of EPCR and other HSC genes in these cells (Figure 2B), we assessed whether they exhibited specific patterns of lineage gene expression by qRT-PCR (Figure 3A). Notably, we found that expression of megakaryocyte (Mk) lineage genes *Gata1* and *Gfi1b* [21] increased significantly in the PU.1^lo^ SLAM cell fraction following IL-1β treatment (Figure 3B). Furthermore, the PU.1^lo^ SLAM fraction became highly enriched for CD41^+^ cells, (Figure 3C,D), indicating that low PU.1 levels enrich for Mk-primed SLAM cells following IL-1β treatment. On the other hand, PU.1^hi^ SLAM cells exhibited robust expression of myeloid lineage and PU.1 target genes, including *Cebpe*, *Cd34*, *Tnfrsf1a*, and *Csfr3 (G-CSFR)*, with the expression of these genes increasing further in response to IL-1β (Figure 3B). Additionally, IL-1β treatment triggered the expression of Mac-1, which is also a PU.1 target gene [6], in a significant proportion of PU.1^hi^ SLAM cells (Figure 3C,E), suggestive of myeloid priming in this compartment. Consistent with these observations, EPCR^−^ SLAM cells, which constitute most of the PU.1^lo^ SLAM fraction following IL-1β treatment, express higher levels of *Gata1* and *Gfi1b*, whereas EPCR^+^ SLAM cells express higher levels of myeloid lineage and/or PU.1 target genes (Appendix A). Thus, low PU.1 levels identify two distinct populations: (1) EPCR^+^ HSC that gain PU.1 expression following IL-1β treatment and (2) EPCR^−^ cells with extensive Mk lineage priming, which do not increase PU.1 expression in response to IL-1β.

Previous work has shown that the SLAM marker CD150 identifies HSC with extensive repopulating capacity that give rise to high levels of myeloid donor chimerism [22,23,24,25], commonly referred to as ‘myeloid-biased’ HSC. Thus, we assessed whether CD150 surface expression levels in the SLAM compartment correspond to PU.1-EYFP reporter activity by quantifying PU.1-EYFP levels in CD150^hi^ and CD150^lo^ SLAM cells (Appendix A). While PU.1-EYFP levels increased in both subsets, they were highest in CD150^lo^ SLAM cells following IL-1β treatment (Appendix A). Likewise, surface expression of Mac-1 was most highly induced by IL-1β in the CD150^lo^ SLAM compartment, while CD41 was significantly expressed exclusively in the CD150^hi^ fraction (Appendix A). Hence, CD150^hi^ does not represent a faithful surrogate marker for high PU.1 expression levels in SLAM cells. We also assessed the extent to which CD150^hi^ enriches for phenotypic EPCR^+^/CD34^−^ HSC^LT^ following IL-1β treatment. Interestingly, while CD150^hi^ does enrich for HSC^LT^ in -IL-1β control mice, it instead enriches for EPCR^−^ cells following IL-1β treatment (Appendix A). Therefore, CD150 does not appear to faithfully report myeloid lineage priming or phenotypic HSC^LT^ in IL-1β-treated mice. Altogether, our data show that PU.1 levels are associated with distinct lineage priming in response to IL-1β, with PU.1^lo^ SLAM cells exhibiting Mk priming, whereas PU.1^hi^ SLAM cells exhibit myeloid lineage priming.

### 3.4. Rapid PU.1 Induction and Expansion of Lineage-Primed SLAM Cells in Response to IL-1β

Given that our studies demonstrate significant phenotypic changes aligned with PU.1 expression in the SLAM compartment under chronic inflammatory stress, we next asked whether these changes are rapidly induced in response to IL-1β or are a result of long-term stimulation. To address this question, we treated PU.1-EYFP mice for 1 day (acute stimulation) and analyzed their hematological parameters (Figure 4A). Like chronic IL-1β stimulation, peripheral blood neutrophil counts increased significantly relative to -IL-1β control mice, yet platelet counts were unchanged (Appendix A). Furthermore, the phenotypic SLAM compartment increased significantly in number, consistent with the induction of transient cell cycle activity and HSC proliferation that we documented previously [7] (Appendix A). As with chronic IL-1β stimulation, the frequency and number of PU.1^hi^ SLAM cells increased following acute IL-1β treatment, with PU.1 induction again confined to EPCR^+^ SLAM fractions (Figure 4B and Appendix A). Additionally, like chronic IL-1β stimulation, the number of PU.1^lo^ SLAM cells remained relatively constant (Appendix A). Consistent with that observation, we noted a rapid depletion of phenotypic EPCR^+^/CD34^−^ HSC^LT^ from the PU.1^lo^ SLAM fraction (Figure 4C,D). This was again accompanied by a significant increase in the proportion of PU.1^lo^ SLAM cells expressing CD41, indicating that these cells rapidly expand and replace phenotypic HSC^LT^ within in the PU.1^lo^ fraction (Figure 4E,F). In contrast to chronic stimulation, Mac-1 surface expression did not increase in the PU.1^hi^ cells, though *Itgam* (*Mac-1*) induction was detectable by qRT-PCR (Appendix A), indicating that expression of the protein is a delayed event. Furthermore, gene expression analyses identified the same pattern of lineage priming in PU.1^lo^ vs. PU.1^hi^ SLAM cells, with PU.1^lo^ cells exhibiting increased Mk lineage gene expression, whereas PU.1^hi^ cells exhibited increased expression of myeloid-lineage-associated genes (Figure 4G and Appendix A). Finally, as with our chronic IL-1β model, we found that the CD150^hi^ marker definition enriches significantly less for EPCR^+^ HSC^LT^ following acute inflammatory challenge (Appendix A). Altogether, our data show that phenotypic remodeling of the SLAM compartment occurs rapidly in response to IL-1β, resulting in the expansion of PU.1^hi^ SLAM cells with myeloid priming and EPCR^−^ PU.1^lo^ SLAM cells with extensive Mk priming.

### 3.5. PU.1^lo^ SLAM Cells from IL-1β-Treated Mice Exhibit Characteristics of SL-MkP

We next interrogated the functional properties of PU.1^hi^ and PU.1^lo^ SLAM cells isolated from mice treated ± IL-1β for 1 day using CFU assays to score lineage potential and clonogenic activity (Figure 5A). Strikingly, we noticed that the PU.1^lo^ SLAM fraction from IL-1β-treated mice exhibited low levels of clonogenic activity relative to controls (Figure 5B), in line with their almost exclusively EPCR^−^ phenotype (Figure 4C). Furthermore, consistent with the Mk gene expression priming in this fraction, of the few colonies we identified, a substantial fraction contained Mk either alone or in combination with myeloid cells (Mix colonies). PU.1^lo^ SLAM cells exhibited an intermediate clonogenic activity, likely owing to this population being a mixture of phenotypic EPCR^+^ HSC^LT^ and EPCR^−^ SLAM cells with Mk priming and limited functional potential. On the other hand, consistent with myeloid priming, PU.1^hi^ SLAM cells exhibited robust granulocyte-macrophage clonogenic activity, producing more colonies than either PU.1^lo^ fraction (Figure 5B). Furthermore, colonies derived from PU.1^hi^ SLAM cells were devoid of Mk, again consistent with their myeloid-primed gene expression phenotype. Collectively, these data show that the colony-forming potential of PU.1^lo^ and PU.1^hi^ SLAM cells aligns closely with their lineage priming.

The lack of clonogenic activity in PU.1^lo^ SLAM cells from IL-1β-treated mice raised the question as to whether this compartment might be enriched for stem-cell-like MkP (SL-MkP) [26]. SL-MkP are Mk-lineage-committed SLAM cells that express high levels of CD41 and rapidly differentiate into Mk, serving as a rapid response mechanism to generate platelets following inflammatory stress. SL-MkP do not possess substantial long-term potential but can rapidly produce Mk in liquid culture, and in many cases, they differentiate into single or few Mk, which would be impossible to identify in a methylcellulose-based CFU assay. To address this possibility, we cultured each SLAM fraction and analyzed their number, morphology, and phenotype (Figure 5C). Notably, we found that both PU.1^lo^ SLAM fractions gave rise to large cells resembling Mk progenitors after 3 days of culture (Figure 5D, see arrows), whereas only a small number of large cells were evident in cultures of PU.1^hi^ SLAM fractions. After four days, we analyzed the cultures by flow cytometry and observed that PU.1^lo^ SLAM cells from IL-1β-treated mice gave rise to a relatively small number of cells versus the other fractions (Appendix A), similar to the limited expansion potential of SL-MkP versus HSC. Approximately 70% of these cells expressed CD41, consistent with an SL-MkP surface phenotype (Figure 5E,F). Notably, PU.1^lo^ SLAM cell cultures from control mice, as well as PU.1^hi^ SLAM cultures from either condition, had a much lower relative abundance of CD41^+^ cells, with lower overall expression of CD41 in the cultures (Figure 5E,F). Taken together, these data support a model in which acute IL-1β stimulation rapidly induces the expansion of PU.1^lo^/EPCR^−^/CD41^+^ cells with characteristics of SL-MkP, which contribute to increased platelet output.

## 4. Discussion

Here, we have used a PU.1-EYFP reporter mouse system to address the dynamics of PU.1 expression in the SLAM compartment under inflammatory stress in vivo. We show that PU.1 expression levels are linked to distinct cell fate outcomes, specifically long-term reconstitution capacity and lineage output. We find that IL-1 triggers numerical expansion of PU.1^hi^ SLAM cells with myeloid lineage priming and demonstrate that the PU.1^lo^ SLAM compartment rapidly becomes enriched for non-repopulating cells with characteristics of SL-MkP following IL-1β stimulation. We link PU.1 expression back to previously defined surface marker systems for prospectively identifying HSC, and find that EPCR expression, but not CD150 expression, can distinguish phenotypic HSC^LT^ from SL-MkP within the PU.1^lo^ fraction. Altogether, our data provide new insights into the molecular and functional heterogeneity contained within the SLAM compartment following inflammatory stress.

Low PU.1 levels are classically associated with long-term HSC activity, as PU.1 activates gene programs that drive myeloid and lymphoid lineage commitment. Indeed, previous studies from our group and others using the PU.1-EYFP reporter system have demonstrated that HSC express low (but not negative) levels of this transcription factor, and that PU.1 is required for normal HSC function, including in inflammatory stress settings. While single-cell tracking studies using the PU.1-EYFP reporter have demonstrated varying expression levels of PU.1 within HSC-enriched cell fractions, the extent to which these differences are linked to functional activities within HSC-enriched populations has to our knowledge not yet been explored. Here, we find that PU.1 levels are in fact associated with functional heterogeneity in the phenotypic SLAM compartment. Interestingly, the PU.1^lo^ SLAM fraction enriches for two distinct activities: long-term repopulating HSC^LT^, which are phenotypically EPCR^+^, and cells with features of SL-MkP, which are phenotypically EPCR^−^, have limited to no repopulating activity and serve as an ‘emergency’ reservoir for platelets in response to inflammatory stress. That both compartments express low levels of PU.1 is perhaps unsurprising given traditional models of cross-antagonism between Gata1 (which regulates Mk/E lineage differentiation) and PU.1 [27,28,29]. However, the prevailing dogma that low PU.1 levels are permissive for high Gata1 expression (and vice versa) has been challenged by recent single-cell resolution fate tracking studies using HSC from mixed PU.1/Gata1 reporter mice [15,17] and single-molecule imaging using smFISH [30]. In both cases, analyses of PU.1 and Gata1 expression patterns have revised canonical model of cross-antagonism, showing that PU.1 and Gata1 can be expressed simultaneously, reversibly and/or in stochastic fashion prior to lineage bifurcation, with Gata1^+^ Mk emerging from HSC initially expressing robust PU.1 levels, for example. While our methodology is unable to resolve fate outcomes for individual HSC clones, we do find that following IL-1β challenge, PU.1 expression increases exclusively within EPCR^+^ SLAM cell fractions that retain multilineage repopulating activity, whereas PU.1 levels remain low in the EPCR^−^ SL-MkPs. Hence, SL-MkPs may be sufficiently Mk lineage committed that *PU.1* has been repressed and cannot be further induced. Given the similarity in PU.1 levels between HSC^LT^ and SL-MkPs under homeostatic conditions, it is tempting to speculate the latter is a direct offshoot from HSC^LT^. Fate tracing studies using inducible reporters driven from inducible drivers that mark functional HSC^LT^ (such as *Fgd5-*, *Tie2-* or *Krt18-Cre*) [31,32,33] and/or single-cell barcoding studies will be needed to rigorously address this point in the setting of inflammatory stress.

In contrast to PU.1^lo^ cells, the PU.1^hi^ SLAM fraction contains cells with intermediate long-term repopulating activity and extensive myeloid priming, which may be reflected in the high level of myeloid clonogenic activity we observe regardless of IL-1β stimulation. A subset of these cells also expresses low levels of Mac-1, a myeloid surface marker often excluded from analysis by the ‘lineage’ antibody cocktail used in flow cytometry analysis of adult (but not fetal) HSPC. Our previous work indicates that Mac-1 is most robustly expressed on EPCR^+^ SLAM cells, including EPCR^+^/CD34^−^ HSC^LT^ [6]. Whether Mac-1 expression in these compartments is indicative of enhanced myeloid priming or functions to maintain HSC lodging in the niche under inflammatory stress remains to be determined empirically [34]. We do note that while evidence of a ‘myeloid bias’ or increased myeloid potential in PU.1^hi^ SLAM cells may be apparent in our clonogenic assays, this is not recapitulated in our transplantation data. We believe two factors are at play. First, increased PU.1 expression following inflammatory stimulation is dynamic. We and others have shown that elevated PU.1 expression is reversible and is not sustained in HSC following the removal of IL-1β or other cytokines [3,9]. Second, the ‘myeloid bias’ phenotype in the setting of transplantation appears closely linked to high levels of HSC activity, and the original characterizations of ‘myeloid-biased’ HSC interpret the elevated myeloid lineage output to be at least in part related to the engraftment of highly potent primitive HSC [24,25]. It is worth noting that high levels of CD150 expression, which is used to mark ‘myeloid-biased’ HSC, does enrich for EPCR^+^/CD34^−^ HSC^LT^ under homeostatic conditions, underscoring the link between myeloid lineage output and enriched HSC activity. However, high CD150 expression instead enriches for EPCR^−^ cells following IL-1β stimulation and thus may not represent a faithful marker for ‘myeloid-biased’ HSC under stress conditions. Here, we find that PU.1^hi^ SLAM cells do contain some phenotypic EPCR^+^/CD34^−^ HSC^LT^; however, their reduced capacity for long-term repopulation indicates that these cells may be in the process of differentiating into MPPs and/or have accumulated divisional history, which restricts long-term potential [35]. Our study is unable to discern whether PU.1^lo^ HSC^LT^ directly feed into the expanded PU.1^hi^ HSC^LT^ compartment following IL-1β treatment or whether this fraction is fed by HSC^LT^ expressing intermediate PU.1 levels. Our previous in vitro single-cell tracking studies showed that IL-1β treatment led to a significant reduction in the lowest PU.1 expressing fraction of cells, which in turn fed into all but the highest (and most rare) PU.1-expressing fraction [7]. Further fate tracking analyses as described above can evaluate the extent to which elevated PU.1 activity corresponds to changes in long-term repopulating activity.

Our study shows that IL-1β triggers the expansion of CD41^+^ SLAM cells that have phenotypic and functional characteristics of SL-MkP [26]. Based on our surface marker data, we would characterize SL-MkP as an EPCR^−^/CD41^+^ SLAM cell fraction, consistent with our prior work showing that EPCR^−^ SLAM cells (regardless of CD34 expression) exhibit relatively high levels of cell cycle activity, robust CD41 expression, and limited long-term repopulating activity following IL-1β treatment. Interestingly, these cells also appear to possess limited clonogenic activity. The initial detailed characterization of SL-MkP demonstrated that individual cells will frequently give rise to only small numbers of Mk or even single Mk, which can be readily detected in liquid culture but not in methylcellulose. These observations also likely explain the intermediate clonogenic activity of PU.1^lo^ SLAM cells from -IL-1β control mice relative to the PU.1^hi^ SLAM cell fraction, as they are probably a mixture of EPCR^+^ HSC^LT^ and EPCR^−^ SL-MkP. SL-MkP are capable of rapid Mk differentiation, typically within a few days if isolated from a mouse challenged with an inflammatory stimulus [26]. Consistent with these observations, we find that PU.1^lo^ SLAM cells give rise to large CD41^+^ Mk-progenitor-like cells within three days. Notably, we do not see rapid generation of such large CD41^+^ cells from PU.1^hi^ SLAM cells. Hence, SL-MkP could be a direct offshoot of PU.1^lo^ HSC^LT^, indicating that PU.1 levels may very well govern this fate outcome. While our study does not formally demonstrate this hierarchy as discussed above, our data are compatible with gene reporter studies and functional analyses showing that Mk markers such as von Willebrand factor (vWF) faithfully mark a primitive subset of HSC capable of robust long-term engraftment and production of Mk [36,37]. Altogether, our data provide further insight into the role of PU.1 in regulating HSC fate outcomes. Linking the PU.1-EYFP reporter to other systems such as *vWF-GFP* reporter mice [36] to track the dynamics of SLAM cell populations under homeostatic stress conditions, and/or modulating PU.1 levels to formally establish its role in governing HSC fate decisions, remain important further studies. Furthermore, as megakaryopoiesis is dysregulated in the contexts of aging [38] and myeloproliferative neoplasms such as essential thrombocytosis and myelofibrosis, understanding the extent to which alterations in PU.1 function contribute to these phenotypes remains an important area of investigation.

## Figures and Tables

**Figure 1 cells-11-00680-f001:**
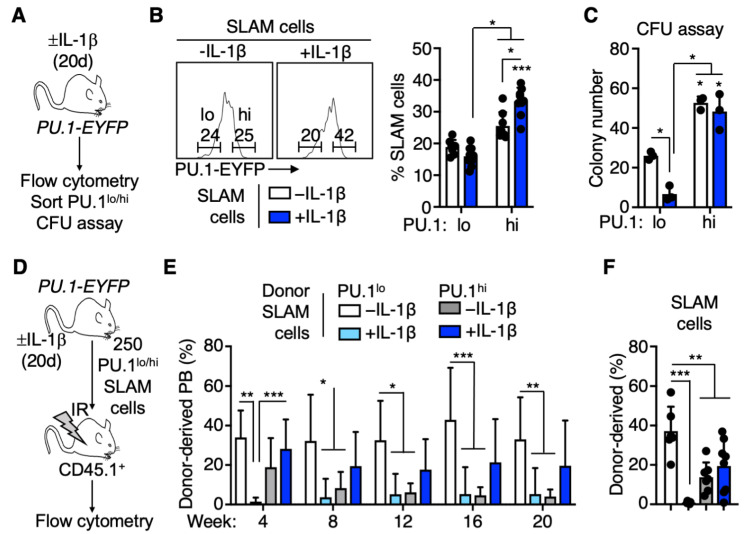
PU.1 levels define distinct repopulating activities following chronic IL-1β treatment. (**A**) Experimental design for isolation and functional characterization of PU.1^hi/lo^ SLAM fractions of PU.1-EYFP mice treated for 20 d IL-1β. (**B**) Representative FACS plot (left) and summary data (right) showing frequency of PU.1^lo^ and PU.1^hi^ SLAM cells in PU.1-EYFP mice treated for 20 d IL-1β (*n* = 6–8/grp). Individual values are shown, and bars represent mean values. Data are compiled from two independent experiments. (**C**) CFU assay of PU.1^lo^ and PU.1^hi^ SLAM cells isolated from PU.1-EYFP mice treated for 20 d IL-1β (*n* = 3/grp). Individual values are shown, and bars represent mean values. Data are representative of two independent experiments. (**D**) Experimental design for isolation and transplant of purified PU.1^hi^ and PU.1^lo^ SLAM cells from PU.1-EYFP mice treated for 20 d IL-1β into lethally irradiated recipient mice. (**E**) Summary data showing donor peripheral blood chimerism in transplanted recipient mice (*n* = 6–8/grp). Data are compiled from two independent experiments. Bars represent mean values. (**F**) Summary data showing donor BM SLAM cell chimerism in recipient mice at 20 weeks post transplant (*n* = 6–8/grp). Individual values are shown, and bars represent mean values. Data are compiled from two independent experiments. Data are shown as means SD. Significance was determined by ANOVA with Tukey’s post-test. * *p* < 0.05; ** *p* < 0.01; *** *p* < 0.001.

**Figure 2 cells-11-00680-f002:**
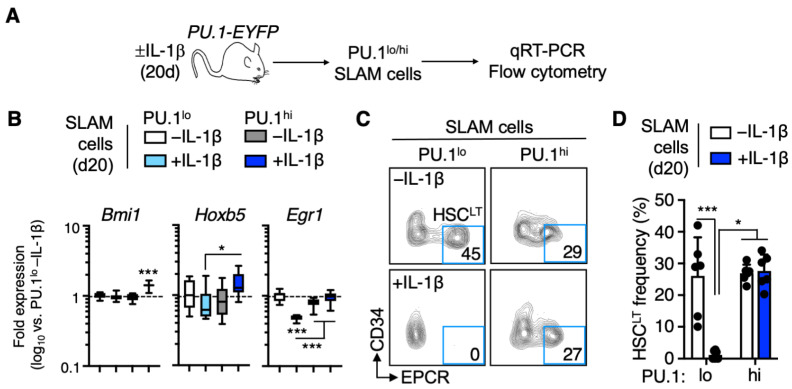
EPCR^+^ HSCLT are depleted from the PU.1^lo^ SLAM fraction following IL-1β treatment. (**A**) Experimental design for analysis of SLAM cells from PU.1-EYFP mice treated for 20 d IL-1β. (**B**) Fluidigm qRT-PCR analysis of HSC genes in PU.1^lo^ and PU.1^hi^ SLAM cells from PU.1-EYFP mice treated for 20 d IL-1β (*n* = 8/grp). Data are expressed as log10 fold change versus PU.1^lo^ -IL-1β. Box represents upper and lower quartiles, with line representing median value. Whiskers represent minimum and maximum values. Data are representative of two independent experiments. (**C**) Representative FACS plot showing phenotypic HSCLT frequencies within the SLAM gate in PU.1-EYFP mice treated for 20 d IL-1β. (**D**) Summary data showing phenotypic HSCLT frequency within the SLAM gate of PU.1-EYFP mice treated for 20 d IL-1β (*n* = 6/grp). Individual values are shown, and bars represent mean values. Data are compiled from two independent experiments. Data are shown as means SD. Significance was determined by ANOVA with Tukey’s post-test. * *p* < 0.05; *** *p* < 0.001.

**Figure 3 cells-11-00680-f003:**
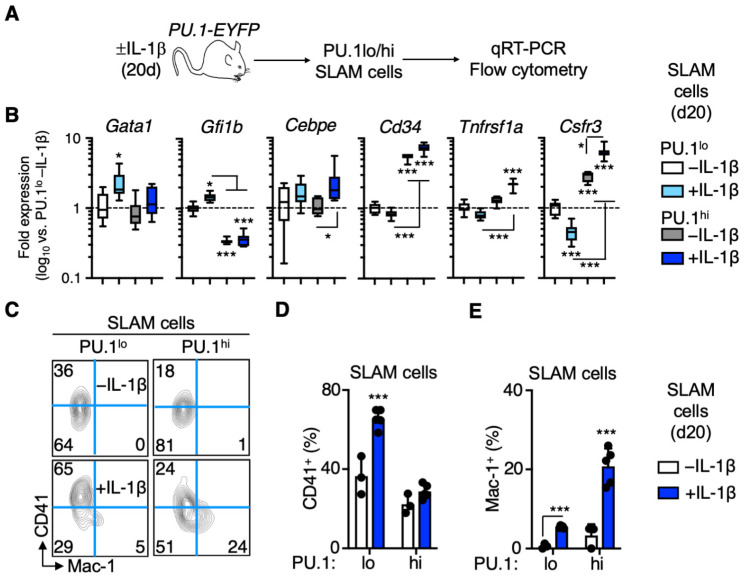
Distinct patterns of lineage priming in the SLAM compartment associated with PU.1 levels. (**A**) Experimental design for analysis of SLAM cells from PU.1-EYFP mice treated for 20 d IL-1β. (**B**) Fluidigm qRT-PCR analysis of megakaryocyte/erythroid and myeloid lineage genes in PU.1^lo^ and PU.1^hi^ SLAM cells from PU.1-EYFP mice treated for 20 d IL-1β (*n* = 8/grp). Data are expressed as log10 fold change versus PU.1^lo^ -IL-1β. Box represents upper and lower quartiles, with line representing median value. Whiskers represent minimum and maximum values. Data are representative of two independent experiments. (**C**) Representative FACS plot showing frequencies of CD41^+^ and Mac-1^+^ cells within the SLAM gate in PU.1-EYFP mice treated for 20 d IL-1β. (**D**) Frequency of CD41+ cells within the SLAM gate of PU.1-EYFP mice treated for 20 d IL-1β (*n* = 3–5/grp). Individual values are shown, and bars represent mean values. Data are representative of two independent experiments. (**E**) Frequency of Mac-1+ cells within the SLAM gate of PU.1-EYFP mice treated for 20 d IL-1β (*n* = 3–5/grp). Individual values are shown, and bars represent mean values. Data are representative of two independent experiments. Data are shown as means SD. Significance was determined by ANOVA with Tukey’s post-test. * *p* < 0.05; *** *p* < 0.001.

**Figure 4 cells-11-00680-f004:**
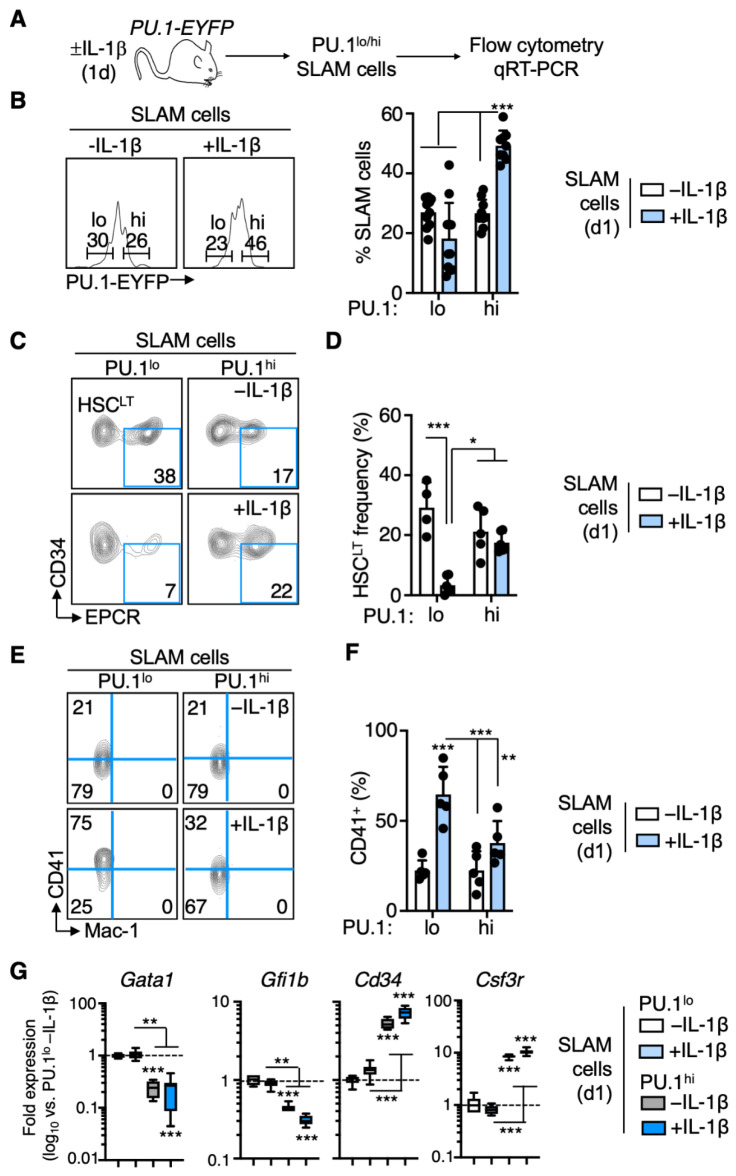
Rapid PU.1 induction and expansion of lineage-primed SLAM cells in response to IL-1β. (**A**) Experimental design for analysis of PU.1-EYFP mice treated for 1 d IL-1 (*n* = 6/grp). (**B**) Representative FACS plot (left) and summary data (right) showing frequency of PU.1^lo^ and PU.1^hi^ SLAM cells in PU.1-EYFP mice treated for 20 d IL-1β (*n* = 8–10/grp). Individual values are shown, and bars represent mean values. Data are compiled from two independent experiments. (**C**) Representative FACS plot showing phenotypic HSCLT frequencies within the SLAM gate in PU.1-EYFP mice treated for 20 d IL-1β. (**D**) Summary data showing phenotypic HSCLT frequency within the SLAM gate of PU.1-EYFP mice treated for 20 d IL-1β (*n* = 4–6/grp). Individual values are shown, and bars represent mean values. Data are compiled from two independent experiments. (**E**) Representative FACS plot showing frequencies of CD41+ and Mac-1+ cells within the SLAM gate in PU.1-EYFP mice treated for 1 d IL-1β. (**F**) Frequency of CD41+ cells within the SLAM gate of PU.1-EYFP mice treated for 1 d IL-1β (*n* = 5/grp). Individual values are shown, and bars represent mean values. Data are representative of two independent experiments. (**G**) Fluidigm qRT-PCR analysis of genes in PU.1^lo^ and PU.1^hi^ SLAM cells from PU.1-EYFP mice treated for 1 d IL-1β (*n* = 8/grp). Data are expressed as log10 fold change versus PU.1^lo^ -IL-1β. Box represents upper and lower quartiles with line representing median value. Whiskers represent minimum and maximum values. Data are representative of two independent experiments. Data are shown as means SD. Significance was determined by ANOVA with Tukey’s post-test. * *p* < 0.05; ** *p* < 0.01; *** *p* < 0.001.

**Figure 5 cells-11-00680-f005:**
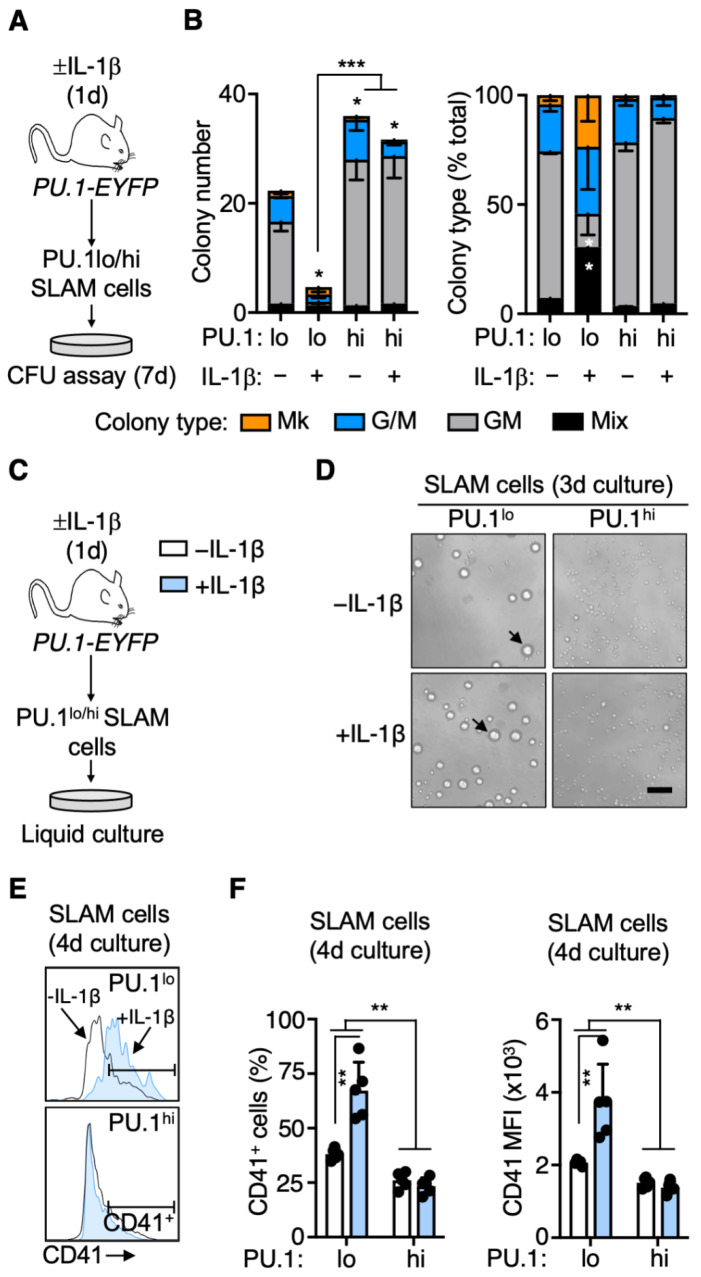
PU.1^lo^ SLAM cells from IL-1β-treated mice exhibit characteristics of SL-MkP. (**A**) Experimental design for in vitro methylcellulose culture analysis of PU.1^lo^ and PU.1^hi^ SLAM cells from PU.1-EYFP mice treated for 1 d IL-1β (*n* = 3/grp). (**B**) Colony number (left) and colony type distribution (right) of methylcellulose-cultured PU.1^lo^ and PU.1^hi^ SLAM cells from PU.1-EYFP mice treated for 1 d IL-1β (*n* = 3/grp). Colony types were based on visual scoring. Colony type distribution is expressed as % total colonies. Data are representative of two independent experiments. Mk: megakaryocytic; G/M. (**C**) Experimental design for in vitro liquid culture analysis of PU.1^lo^ and PU.1^hi^ SLAM cells from PU.1-EYFP mice treated for 1 d IL-1β (*n* = 5/grp). (**D**) Representative phase-contrast microscope images after 3-day culture of PU.1^lo^ and PU.1^hi^ SLAM cells from PU.1-EYFP mice treated for 1 d IL-1β. Scale bar: 100μm. Arrows indicate Mk-progenitor-like cells. (**E**) Representative histograms of CD41 expression after 4-day culture of PU.1^lo^ and PU.1^hi^ SLAM cells from PU.1-EYFP mice treated for 1 d IL-1β. (**F**) Summary data showing frequency of CD41+ cells (left) and overall CD41 expression levels (right) after 4-day culture of PU.1^lo^ and PU.1^hi^ SLAM cells from PU.1-EYFP mice treated for 1 d IL-1β (*n* = 5/grp). Expression data are geometric mean fluorescence intensity (MFI). Individual values are shown, and bars represent mean values. Data are representative of two independent experiments. Data are shown as means SD. Significance was determined by ANOVA with Tukey’s post-test. * *p* < 0.05; ** *p* < 0.01. *** *p* < 0.001.

## Data Availability

The data are available upon request from the corresponding author.

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
