# Peer review of "PU.1 Expression Defines Distinct Functional Activities in the Phenotypic HSC Compartment of a Murine Inflammatory Stress Model"

_cells, 2022, doi:10.3390/cells11040680_

Round 1
Reviewer 1 Report
The manuscript by Chavez et al. described the phenotypic and functional heterogeneity of PU.1 expression following inflammatory challenge with IL-1 stimulation. Using PU.1 reporter mice with either long or short stimulation with IL1, they focus their attention on changes within the HSC with regards to surface marker expression, targeted gene expression or functionality following transplantation (standard for HSC biology). They identify PUlo EPCR- CD41+ Slam+ HSC as an expanded population in response to IL1 stimulation and this occurs rapidly upon IL1 exposure (seen at 24 hours and at 20 days). They profile PUlo HSCs following IL1 stimulation and find that they are enriched in Mk biased transcripts, also supporting their idea. Functionally, PUlo HSC in steady-state or PUhi HSC following IL1 stimulation were the “best” in transplantation assays with long-term reconstitution. Overall, this is a sound body of work, the paper is well written, and the experiments are well-performed. However, there are some points that should be addressed:
Comments to consider and/or address:
- It was difficult to read this paper without constantly referencing the JEM paper also published by this group doing similar experiments and profiling of HSCs. (The Myc-GFP mice in SupplFig2 are a perfect example of this, why is this here, what does it have to do with the science discussed?).
- Knowing that PU.1 is downregulated during Mk/Ery differentiation, wouldn’t you predict that CD41+ HSC don’t express PU.1? I’m not sure of the innovation here correlating CD41 to PU.1. I think the finding that there are more CD41+ HSC with IL1 is new, but that’s in the absence of PU.1 expression.
- Perhaps the most important point, other groups don’t have PU.1 reporter mice – can the combination of CD41, EPCR, CD34 and Mac1 be used to find PU.1hi/lo HSCs without this reporter strain following IL1 stimulation?
- It would be good to know the expression of IL1Rs across these HSC subsets? Also, can we see absolute counts rather than just frequencies of HSC, are these expanding populations over 20 days or just selection of subsets?
- Mac1 expression in adult HSC (not fetal) is interesting. As this is an integrin, does it actually correlate with myeloid skewing, is there expression of corresponding transcription factors that mediate myeloid commitment. Also does this correspond to CD150hi Myl-HSC bias seen in aging?
- The targeted transcripts were done in total PUlo and PUhi, are these biases also seen within EPCR+/- CD34- subsets of PU expression?
- Figure 4B – what are the arrows referencing?
- Figure 4E – somewhat confusing as there are hardly any colonies with PUlo IL1 stimulation relative to PUlo control, is the potential for Mk actually the same between these two groups in terms of Mk numbers (not %)?
- The discussion suggests that because of the PUlo CD41+ HSCs there is the potential to rapidly make platelets, doing a CBC could confirm platelets in the blood?
- If the authors could spell out in the discussion what PU.1 expression specifically tells them that CD41, EPCR and CD34 didn’t already within the context of HSC differentiation potential and function, that would be appreciated.
Author Response
Please see response to comments in attached document.

Reviewer 2 Report
In this manuscript, Chavez and colleagues used functional assays, phenotypic analyses, and molecular profiling to resolve the relationship between PU.1 levels and HSC activity within the SLAM gate. I found the experiments to be thorough, well-controlled, and elegant. The conclusions drawn are supported by their data and the story is well-presented. I have only a few concerns, outlined below, focused on some minor discrepancies in data presentation and interpretation.
- Figure 1E- If the percent of PU.1 high cells increases upon IL1 treatment, but the number of PU.1 low cells remains constant, what population feeds the increase in PU.1 high cells? Is this from the intermediate population? And can you then say that it remains normally distributed?
- Figure 2B-D- The authors state that “the PU.1lo SLAM fraction from PBS-treated mice contained abundant phenotypic EPCR+/CD34- HSCLT relative to PU.1hi cells, consistent with the superior long-term engraftment of PU.1lo SLAM cells from PBS-treated mice”. However, this does not seem to be reflected in the data, as there are no significant differences in the PU.1 lo and PU.1 hi SLAM fractions in PBS-treated mice.
- Figure 3- Were platelet counts done at the 24 hour timepoint? This information would be interesting to determine if the increase in CD41+ HSCs is contributing to functional MKs and platelet production. We often assume this to be true, but the time course of maturation of these CD41+ HSCs is unclear and there are subpopulations of MKs that may not make platelets.
- Figure 4- In general, the images and data in Figure 4 support increased CD41+ cells in the SLAM compartment. However, as CD41 is a very early marker of MK differentiation, I would still refer to this population as MKs and MK progenitors. To measure mature MKs, you would have to look at a marker like CD42b. Another indication of mature MKs is a polyploid nucleus, which could be measured by DNA content in flow cytometry, or seen in images taken of the MKs at a higher magnification.
Minor comments
- S1A- Labels on the x-axis are cut off
- The font styles and sizes in the figure legends are inconsistent
Author Response
We have included our response to reviewer 2 in the document attached.

Reviewer 3 Report
Chavez et al. extend their previous work (REF #6). They take further advantage of a PU.1-eYFP reporter mouse to determine whether there is functional heterogeneity within the phenotypic HSC compartment.
i.e. Do HSCs that express the reporter at low vs high levels exhibit measurable differences. The short answer is yes, and the authors report what they recorded here.
My overall impression is that they dump on the reader a lot of data, but could have taken more care to present their results and interpretations in a narrative that is easier to follow and digest (eg. for trainees).
Furthermore, their interpretations tend to be one sided and alternative interpretations could be discussed more.
For example: line 150 - "On the other hand, Ifitm1 and Foxo3, as well as Cpt1a, were increased in IL-1-exposed PU.1hi SLAM cells, suggesting alterations in metabolic and transcriptional activity that may be related to enforcement of quiescence and/or adaptation to an inflammatory environment."
1) This sentence is in a very long paragraph with the title heading: "The PU.1lo SLAM fraction enriches for HSC-like Mk progenitors following IL-1 exposure." Thus, discussing PU.1 hi cells in this paragraph disrupts coherence and could be divided into 2 paragraphs for instance.
2) It assumes that readers are already familiar with Ifitm1, Foxo3, and Cpt1a. What is the rationale for checking expression of Ifitm1, an IFN-inducible gene? What is the significance? Foxo3 mRNA might be elevated but its protein may not be in the nucleus and thus not functionally relevant. Pardon my ignorance, but I don't know anything about Cpt1a.
3) "suggesting alterations in metabolic and transcriptional activity that may be related to enforcement of quiescence and/or adaptation to an inflammatory environment" -- is too vague and speculative. Authors should explain one by one (each gene, each point) and cite references.
This is just one example but there are others. Did the authors ask any colleagues to read their manuscript?
I'm a fan of graphical abstracts or models (as a last figure or supplementary) that summarize the paper's findings. This would be one suggestion.
Although, this PU.1-eYFP reporter mouse has been "extensively validated," have the authors validated the correlation with PU.1 protein levels under inflammatory conditions? It's formally possible that the reporter might not be 100% faithful under a different condition. Throughout the text, the authors equate PU.1-eYFP levels with PU.1 protein levels which is an assumption (eg. lines 16, 19, 20 in Abstract). If it's not known, then it might be best to spell out PU.1-eYFP each time or add a disclaimer that the authors are assuming PU.1-eYFP levels equate to PU.1 protein levels and/or functional activity.
The Results section (line 67) starts abruptly and may benefit from a bit more pertinent background. How does IL1 affect "SLAM cells"? Using Morrison's SLAM scheme, do the FACS profiles ± IL1b look the same? Absolute cell numbers in bone marrow and subsets thereof? Is IL-1b acting on HSCs directly (ie. do "SLAM cells" and/or LT-HSCs express IL-1 receptor)?
Minor comments:
Lines 24-6: "similar PU.1 transcription factor levels can be tied to distinct functional activities under steady-state and inflammatory conditions." I found this last phrase in Abstract hard to understand.
Line 102: "Figure 20. d" should be "for 20 d"
Fig 1J -- RT-qPCR for Cd34 mRNA doesn't seem to reflect the surface protein levels depicted in Fig 1B. Also see Fig 2L vs 2E. Would be good to check MFI of CD34 staining.
Line 131: " This effect was likely due to IL-1-induced increases in PU.1 expression that shifts HSCLT out of the phenotypic PU.1lo gate6 (Fig. 1C)." I don't understand this explanation and did they mean to refer to Fig 1C or another figure?
Lines 284-7: " Concurrently, we observe depletion of phenotypic EPCR+/CD34- HSCLT in PU.1lo SLAM cells. These findings are consistent with our published observation that PU.1 is robustly induced in phenotypic HSCLT, which removes these cells from the PU.1lo SLAM gate6." In both sentences, authors should add "under inflammatory conditions" to avoid misunderstanding.
Line 317: "Vwf-GFP" comes out of nowhere. Better provide an introduction to it
The authors use IL-1 throughout the text, but did they test that IL-1a and IL-1b have similar effects? If not, then it might be best to spell out IL-1b each time.
Fig 4F -- My overall impression is that IL-1b treatment reduces the colony numbers dramatically in PU.1 lo SLAM cells. Was this discussed?
Fig S1B -- I was surprised that PU.1 hi SLAM cells resulted in such high lymphoid donor chimerism. Isn't high PU.1 expected to promote myeloid lineage?
Fig S2A -- GFP-Myc mice were not described in M & M
Fig S2B -- not useful unless there's statistical analysis
Fig S3A -- I'm not sure what can be concluded from this in vitro liquid culture experiment. " In line with their reduced repopulating activity following transplant and reduced phenotypic HSCLT content, PU.1lo SLAM cells isolated from IL-1-exposed mice also exhibited significantly reduced expansion in liquid culture (Fig. S3A)." However, by days 8 and 12, it appears that they have largely recovered?
The Materials & Methods had a few issues:
- what is genetic background of PU.1-EYFP mice? Is it also C57BL6/J? What age and sex were used in experiments?
- line 383: "microbreads" typo
- line 402: Mac-1 PE/Cy7 has no catalog #
- line 408: no catalog # for c-Kit APC/Cy7, Gr1 PB, CD41 BV510, CD16/32 BV711, CD150 BV785
- lines 413-15: multiple usage of strange spiral symbol
- line 455: how was Gusb chosen for normalization? Was it validated in some way?
Author Response
We have included our response to reviewer 3 in the document attached.
